# Microbiota and Postmenopause: The resilience of intestinal bacteria in the face of female hormonal aging

**Thayane Christine Alves da Silva**[1,2], **Jennefer Aparecida Gonçalves Oliveira**[1,2], **Lauro Ângelo Gonçalves de Moraes**[3], **Izinara Rosse**[3,4], **Aristóteles Góes-Neto**[5], **Vasco Ariston de Carvalho Azevedo**[6], **Renata Guerra-Sá**[1,2]*

**1** Graduate Program in Biotechnology, Biological Sciences Research Center, Federal University of Ouro Preto, Ouro Preto, Minas Gerais, Brazil, **2** Laboratory of Biochemistry and Molecular Biology, Department of Biological Sciences, Institute of Exact and Biological Sciences, Federal University of Ouro Preto, Ouro Preto, Minas Gerais, Brazil, **3** Bioinformatics Multiusers Laboratory, Federal University of Ouro Preto, Ouro Preto, Minas Gerais, Brazil, **4** Department of Pharmacy, School of Pharmacy, Federal University of Ouro Preto, Ouro Preto, Minas Gerais, Brazil, **5** Laboratory of Molecular and Computational Biology of Fungi, Department of Microbiology, Institute of Biological Sciences, Federal University of Minas Gerais, Belo Horizonte, Minas Gerais, Brazil, **6** Department of Genetics, Ecology and Evolution, Federal University of Minas Gerais, Belo Horizonte, Minas Gerais, Brazil

* rguerra@ufop.edu.br

## Abstract

### Background

Female aging is characterized by a decline in ovarian function until the establishment of the post-menopausal state. Other research lines seek to understand how the gut microbiota contributes to these diseases at all stages of life, including post-menopause, and its potential to be used as an ally to promote healthy aging and as a biomarker that can be associated with the pre-and post-menopausal period.

### Methods

Post-menopausal women (n = 44) aged between 45 and 60, divided into Group A n = 34 (up to 10 years postmenopause) and Group B n = 10 (more than 10 years postmenopause) had fecal samples analyzed by metabarcoding using the 16S rRNA gene, and bacterial composition and diversity were compared between the two groups.

### Results

Both groups showed high diversity in microbiota according to Shannon's index but with no difference. Simpson's index indicated that post-menopausal women over 10 had a more diverse microbiota with lower species dominance (p-value 0.04). The predominant organisms were Phyla Firmicutes, Bacteroidota, Proteobacteria, and Actinobacteria; Families *Lachnospiraceae, Bacteroidaceae, Ruminococcaceae,*

**Data availability statement:** The datasets used and/or analyzed during the current study are available from the corresponding author and also in a public domain repository https://github.com/bioinfonupeb/redemicro-thayane and the data underlying the results presented in the study are available from the National Center for Biotechnology Information (NCBI, https://www.ncbi.nlm.nih.gov/) under the BioProject ID PRJNA1223961.

**Funding:** This research is supported by CAPES (Coordenação de Aperfeiçoamento de Pessoal de Nível Superior), CNPQ (Conselho Nacional de Desenvolvimento Científico e Tecnológico), FAPEMIG (Fundação de Amparo à Pesquisa do Estado de Minas Gerais), RED-00132-16 and RED-001181-23 and the Brazilian National Council for Scientific and Technological Development.The funders had no role in the study design, data collection and analysis, decision to publish or preparation of the manuscript.

**Competing interests:** The authors have declared that no competing interests exist.

**Abbreviations:** AMH, Anti-Mulerian hormone; ASVs, Amplicon Sequence Variants; Bp, Base pairs; DNA, Deoxyribonucleic acid; FSH, Follicle-stimulating hormone; IHMS, International Human Microbiome Standards; LBBM, Biochemistry and Molecular Biology Laboratory; LH, Luteinizing hormone; OTUs, Operating taxonomic units; PCoA, Principal coordinates analysis; PERMANOVA, permutational multivariate analysis of variance; Phred, Phil's Read Editor; Qiime 2, Quantitative Insights into Microbial Ecology; UFOP, Federal University of Ouro Preto.

*Prevotellaceae;* Genera *Bacteroides, Prevotella, Faecalibacterium, Agathobacter* and *Dialister.* There were no differences between the groups.

## Conclusion

Women who had been post-menopausal for ten years or more had microbiota with greater diversity and less dominance of taxa. There was no difference between the ten most enriched taxa in each group. Our results indicated that the fecal Microbiota of these women showed a uniform and equitable distribution of the organisms inferred for the groups, regardless of the time elapsed since postmenopause.

## Introduction

A decline in ovarian function characterizes female aging until the post-menopausal state is established. The end of a woman's reproductive phase is evidenced by menopause, when there is a complete depletion of ovarian follicles and, consequently, a decline in hormone levels of estrogen and progesterone [1–7]. Every year, 1.5 million women experience menopause [8]. Those same women will remain under post-menopausal conditions for the rest of their lives, which, besides hormonal changes, include the manifestation of psychological, somatovegetative, and urogenital symptoms [9,10], which can persist for 10 years or more after the menopause [11,12].

Other metabolic disorders are known to have some level of correlation with the occurrence of menopause, such as metabolic syndrome [13,14], dyslipidemia [15], osteoporosis [16,17], Diabetes Mellitus [18,19], and cardiovascular diseases. Other research lines have sought to understand how the intestinal microbiota contributes to these diseases at all stages of life [20–22], including post-menopause [23–26], to minimize or act preventively on women's health in their senescence. It is estimated that the microbial cells that inhabit the human gut exist in a ratio of approximately 1:1 with human cells, while microbial genes outnumber human genes [27]. All this bacterial content, its genes, metabolites, and signaling molecules produced, have the potential to be used as allies to promote healthy aging and/or as biomarkers that can be associated with the pre-and post-menopausal period as predictors of diagnosis, prognosis, and in the monitoring of treatments used for various conditions associated with female aging and its consequences.

This study used a metabarcoding approach to identify the composition and diversity of the intestinal microbiota of brazilian women according to their post-menopausal period to understand the impact that the intestinal microbiota has on human health and its importance.

### Materials and methods

#### 1. Characteristics of the participants

This study was carried out in Ouro Preto – MG, Brazil, in which 44 women took part, ranging in age from 45 to 60 years, were post-menopausal in terms of their reproductive aging stage according to the criteria proposed by STRAW+10 [5]. Participants

who reported an absence of menstrual cycles for more than 12 consecutive months and had endocrine and hormonal changes were included in the study. All participants reported a natural menopause. Women taking medication for glycemic control, antihypertensive, and lipid-lowering drugs were included. Women taking hormone replacement therapy and antibiotics did not take part in the study. Total or partial hysterectomy and bilateral oophorectomy were also considered exclusion criteria. For comparative analyses of the intestinal microbiota , the participants were divided into two groups according to how long they had been post-menopausal: Group A n = 34 (Up to 10 years post-menopausal); Group B n = 10 (More than 10 years post-

menopausal). Participant recruitment and sample collection began on December 15, 2020, and ended on March 5, 2021.

## 2. Ethics approval and consent to participate

This research was approved by the Ethics Committee for research involving human beings at the Federal University of Ouro Preto (UFOP) (protocol number 29723420.9.0000.5150). The entire process of selecting participants, sampling, collection, and subsequent processes was carried out using the standardized protocols recommended by the aforementioned committee. The participating women were informed of their participation and rights and provided oral and written consent for collecting and storing biological samples for later analyses. All procedures were performed according to the relevant guidelines and regulations.

## 3. Gut microbiota analyses

**3.1. Fecal sample collection.** The participants were instructed in the procedures for self-collection and storage of the samples in refrigeration until the project team collected them. Subsequently, the samples were sent to the Biochemistry and Molecular Biology Laboratory (LBBM-UFOP) and stored at −80°C for approximately one month until DNA extraction.

**3.2. DNA extraction.** Bacterial genomic DNA was extracted using 15 mg of feces, following the International Human Microbiome Standards (IHMS) recommendations, specifically protocol H [28]. Briefly, the extraction process used 15 mg of feces, which proceeded with chemical and mechanical lysis steps using guanidine isothiocyanate, N-lauryl Sarcosine, glass beads, and vortex processes. The protocol used Polyvinylpolypyrrolidone and TENP (Tris-HCL buffer, EDTA, NaCl, and PVPP) between centrifugation and washing steps. Finally, the DNA was precipitated with Isopropanol and treated with 10 mg/μL of Rnase (Sigma). The DNA was resuspended in 200 μL of RNAse-free water and stored at 4°C. The NanoDrop (Thermo Fisher Scientific) was used to assess the concentration and purity of the extracted DNA, and agarose gel electrophoresis was used to check its integrity [29].

**3.3. 16S rRNA gene sequencing.** All 44 samples were subjected to high-throughput sequencing using the V3/V4 hypervariable regions of the 16S rRNA gene. For 38 samples, the libraries were made by *Neoprospecta Microbiome Technologies* (Florianópolis, SC, Brazil), using primers 341F (CCTACGGGRSGCAGCAG) and 806R (GGACTACHVGGGTWTCTAAT) [30,31] with the MiSeq Reagent Kit V2 (500 cycles) and the libraries were sequenced (*paired-end*) on the MiSeq Sequencing System platform (Illumina Inc., USA). The primers complement the target and the Illumina adapter, allowing a second PCR by adding the indexing sequences [30]. Two μl of DNA were used for each sample as a template in the first PCR reaction. Platinum Taq (Invitrogen, USA) was used under the following conditions: 95 °C for 5 min, 25 cycles of 95 °C for 45s, 55 °C for 30s and 72 °C for 45s, and a final extension of 72 °C for 2 min for the first PCR. For the second PCR, the conditions were 95 °C for 5 min, 10 cycles of 95 °C for 45s, 66 °C for 30s and 72 °C for 45s, and a final extension of 72 °C for 2 min. The final reactions were purified using Neobeads ® (Sera-Mag ™ based magnetic beads), and an equivalent volume of each sample was added to the sequencing pool. Negative control was used for each PCR, and the reactions were carried out in triplicate. The final DNA concentration of the library pool was estimated using Picogreen dsDNA (Invitrogen, USA) and diluted for qPCR using the Collibri™ Library Quantification Kit (Invitrogen, USA) already optimized for Illumina libraries. The sequencing pool was adjusted to a final concentration of 11 pM. The remaining six samples were sequenced by *Proteimax Biotecnologia Ltda* (São Paulo, Brazil), which used the Illumina Nextera XT DNA library prep kit for the libraries and the primer 314F (CCTACGGGNGGCWGCAG) on the Illumina NovaSeq6000 (2x250 bp) (*single-end*) [30,31].

**3.4. Taxonomic data analysis.** The raw data from sequencing the V3/V4 regions of the 16S rRNA gene was submitted to a metagenomic analysis *pipeline* (Public repository at: https://github.com/bioinfonupeb/redemicro-thayane). In this process, the primer sequences of the *forward* and *reverse* reads were identified by alignment, with a minimum overlap of 8 base pairs (bp), and subsequently removed. Afterward, quality control was carried out using the DADA2 plugin of the QIIME2 (https://qiime2.org/) (*Quantitative Insights Into Microbial Ecology*) software [32]. Only reads with Phred score values greater than 20 and a minimum size of 70 bp were selected. The forward and reverse reads were then joined, followed by the inference of possible sequencing errors and the elimination of chimeric sequences. The entire project was implemented in Python, using plugins available in the QIIME2 microbiome analysis package [32]. In the initial stages of the pipeline, paired-end samples and single-end samples were processed separately. Thus, the samples were divided into two groups, going through all data quality control procedures and independently constructing the ASVs (*Amplicon Sequence Variants*) tables. Once we obtained the ASV tables, they were merged into a single ASV table, which was then used in the subsequent stages of the pipeline. In an attempt to obtain the highest possible percentage of concatenated reads (in the case of paired-end sequencing), different tests were carried out, altering the size of the overlap required between the pairs (we varied the size of the overlap from 1 to 20 and, in the end, used 12) to merge the reads. In this way, the percentages achieved are as high as possible. The next step was the taxonomic inference based on a native Bayesian classification method [35] using the SILVA 16S 132 reference library [36]. The sequencing data is deposited at NCBI under the BioProject ID PRJNA1223961 (http://www.ncbi.nlm.nih.gov/bioproject/1223961).

**3.5. Diversity metrics and statistical analysis.** Taxonomic data was filtered to remove readings below 0.05% (relative abundance) and normalized using total sum normalization (TSS). Alpha diversity was assessed using Shannon and Simpson índices [33–36], and differences between groups were tested using the Kruskal-Wallis test ($p < 0.05$). Beta diversity was assessed using principal coordinates analysis (PCoA), based on Bray-Curtis similarity [35–38] to visualize differences in the overall composition of the Microbiota (β diversity) between groups, jointly with permutational multivariate analysis of variance (PERMANOVA) and analysis of similarities (ANOSIM) test. The Benjamini-Hochberg correction method was used to adjust the p-values. The 10 most enriched/abundant organisms were identified, and the significant difference of these taxa between the groups was assessed by the Mann-Whitney (U) test ($p < 0.05$).

## Results

### 1. Clinical and demographic characteristics of study subjects

Forty-four post-menopausal women classified according to their stage of reproductive aging by the STRAW+10 criteria [5] took part in the study. They were divided into two groups to compare their gut microbiota according to how long they had been post-menopausal: Group A n = 34 (≤10 years post-menopausal) and Group B n = 10 (≥10 years post-menopausal). The average age of the women was 54.7 years, and they had been post-menopausal for 4.82 ± 2.68 years (Group A). Group B was 57.3, 14.2 ± 5.37 years post-menopausal (Table 1). For Groups A and B, respectively, diseases were predominant, such as hypertension (38% and 30%), thyroid disease (29.4% and 30%), and neuropsychiatric disorders (26.5% and 40%). In both groups, the women reported using one or more drugs, with antihypertensive, antidepressant/anxiolytic, hypoglycemic, and thyroid drugs being the most commonly used classes. All the data cited is described in detail in Table 1.

### 2. Sequencing output and diversity estimates

The sequencing was carried out so that 06 samples were sequenced with only *forward* reads (*single-end*), and 38 samples were sequenced in *forward* and *reverse* (*paired-end*). After quality processing, the single-end samples had more than 70% of the reads. In comparison, the *paired-end* samples had between 40 and 76% of the reads that were concatenated, generating a total input of 6,091,976 reads, which, after quality control and joining the data, left 3,747,166 reads (61.5%) that met the recommended quality standards (Table 2). To carefully assess the sequencing quality, we present the rarefaction curves for the 44 samples, the results of which were satisfactory for capturing the taxonomic diversity present in the samples evaluated (Fig 1).

**Table 1. Sociodemographic and health characteristics of the women participating in this study.**

| Characteristics | Group A (n = 34) (≤ 10 Years postmenopause) | Group B (n = 10) (≥ 10 Years postmenopause) |
|---|---|---|
| **Age (years)** | 54.7 ± 3.26 | 57.3 ± 2.86 |
| **Last menstrual cycle (years)** | 50.6 ± 3.14 | 43.8 ± 5.88 |
| **Post-menopause time (years)** | 4.82 ± 2.68 | 14.2 ± 5.37 |
| **BMI (kg/m²)** | 30.90 ± 6.30 | 26.98 ± 5.04 |
| **Education** | | |
| 0 to 8 years | 45% | 50% |
| More than 8 years | 55% | 50% |
| **Women's income** | | |
| <1 Salary | 17.6 | 60.0 |
| 1 Salary | 47.1 | 0 |
| 1-2 Salaries | 20.6 | 20.0 |
| >5 Salaries | 2.9 | 10.0 |
| NA | 11.8 | 10.0 |
| **Income Family** | | |
| <1 Salary | 2.9 | 40.0 |
| 1 Salary | 11.8 | 0 |
| 1-2 Salaries | 44.1 | 20.0 |
| 3-5 Salaries | 26.5 | 20.0 |
| >5 Salaries | 2.9 | 10.0 |
| NA | 11.8 | 10.0 |
| **Hypertension** | 38% | 30% |
| **Diabetes** | 20.6% | 20% |
| **Osteoporosis** | 11.8% | 20% |
| **Thyroid diseases** | 29.4% | 30.0% |
| **Neuro-psychiatric (Anxiety, Depression, or both)** | 26.5% | 40% |
| **Use of drugs** | 64.7% | 60.0% |
| Antihypertensive | 44.1% | 4.5% |
| Antidiabetic | 20.5 | 4.5 |
| Lipid-lowering agents | 8.8 | 2.2 |
| Antidepressant/anxiolytic | 26.4 | 6.82 |
| Thyroid agents | 23.5 | 4.5 |
| Analgesic/anti-inflammatory | 17.6 | 0 |
| Bone diseases | 2.9 | 2.2 |
| Antiulcer | 5.8 | 0 |
| Bronchodilators | 5.8 | 2.2 |
| Vitamins/supplements | 5.8 | 0 |
| Drugs for insomnia | 5.8 | 0 |
| Others | 2.9 | 2.2 |
| **Medication name** | Acetylsalicylic acid, Alenia, Alprazolam, Amitriptyline, Amlodipine, Atenolol, Atorvastatin, Betahistine, Budesonide, Buspirone, Calcium, Chlorthalidone, Clonazepam, Enalapril, Ezetimibe, Fluoxetine, Formoterol, Furosemide, Glibendamycin, Hydrochlorothiazide, Levothyroxine, Losartan, Metformin, Nimesulide, Omeprazole, Pantoprazole, Sertralin, Simvastatin, Tandrilax, Venlafaxine, Vitamin B12, Zoloft, Zolpidem | Alendronate, Atenolol, Atorvastatin, Clonazepam, Dulexetin, Flunarizine, Hydrochlorothiazide, Levothyroxine, Losartan, Metformin, Pregabalin, Thiamazole, Venlafaxine |

*(Continued)*

**Table 1.** (Continued)

Information is based on the women's report of a medical doctor's previous diagnosis: BMI (Body mass index), Other (Betaistin, Flunarizine). NA (Did not answer). The minimum wage in Brazil in 2021 is USD 206,49. Participants reported using these medications during their participation in this study and reported using more than one class of drugs concomitantly. Elaborated by the author.

**Table 2.** Data inferred after quality control and merging reads were obtained for single-end and paired-end.

| Samples | Total *Input* | *Input* after quality control |
|---|---|---|
| Single-end (n=06) | 473103 (100%) | 399704 (84,4%) |
| Paired-end (n=38) | 5618873 (100%) | 3347462 (59,5%) |
| Concatenated *reads* (single-end/ paired-end) | 6091976 (100%) | 3747166 (61,5%) |

Reads concatenated in the quality control process and used for taxonomic inference. Elaborated by the author.

**Fig 1. Rarefaction curve of the features observed per sample analyzed (n=44).** The sequencing was satisfactory for representing bacterial diversity since the rarefaction curves reached the Plateau as shown in the figure. Elaborated by the author.

## 3. Microbiota diversity

**3.1. Alpha-diversity.** In this study, the richness and diversity of the gut microbiota of post-menopausal women were assessed according to their length of postmenopause. The Shannon index (Fig 2A) showed that both groups had a high diversity of species in their intestinal microbiota but with no significant difference when compared (p-value: 0.07). Simpson's index (Fig 2B) showed that Group B was statistically different from Group A (p-value: 0.04), indicating that the bacterial community in this group had greater diversity with lower taxon dominance. The results for Alpha diversity found in the cohort of this study showed similarity between the organisms found in the fecal microbiota of those women and a uniform and equitable distribution of the microorganisms inferred for the groups regardless of the time elapsed since postmenopause.

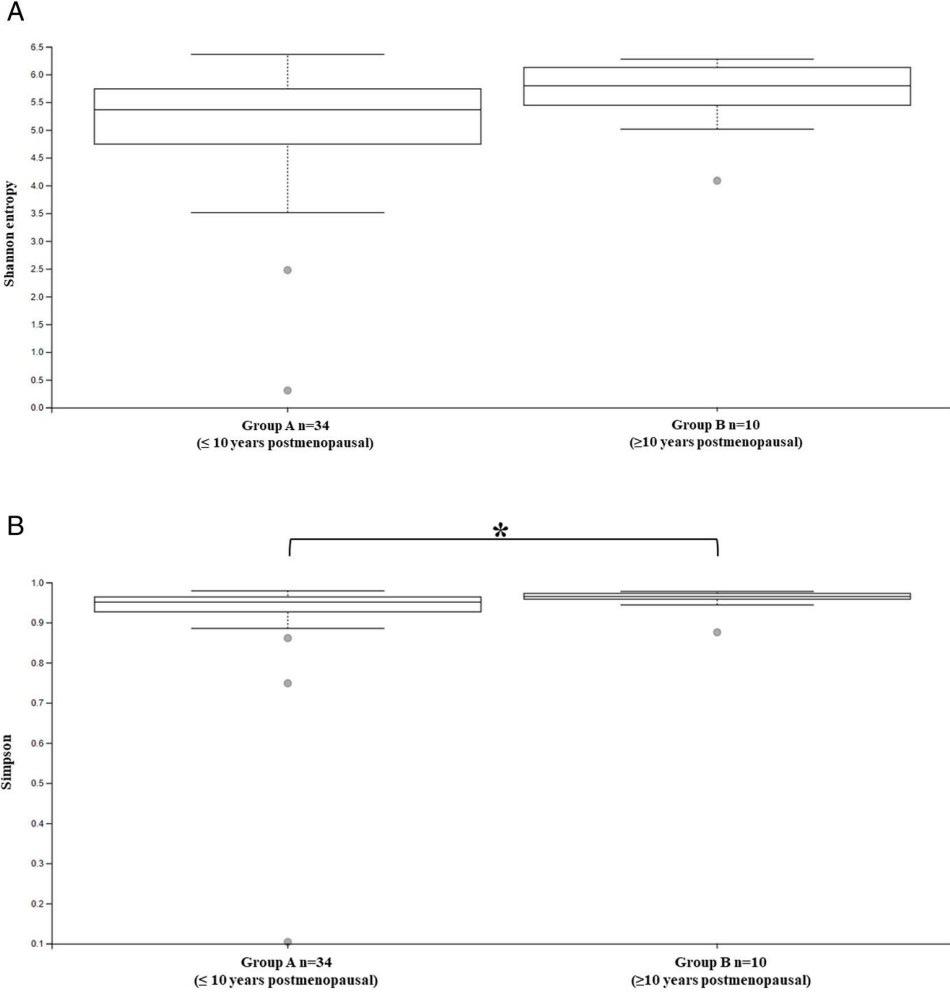

**Fig 2. Alpha diversity of the fecal microbiota presented by the Shannon and Simpson index. A)** Shannon index (p-value: 0.07); **B)** Simpson index (p-value: 0.04). Legend: Group A n = 34 (≤10 years postmenopause); Group B n = 10 (≥10 years postmenopause). The Kruskal-Wallis statistical test was used for all indices: **A)** Shannon (p-value: 0.07); **B)** Simpson (p-value: 0.04) significant differences found by Simpson's index. Elaborated by the author.

### 3.2. Beta diversity.
The principal coordinates analysis (PcoA) (Fig 3) showed proximity and similarity between the bacterial groups in groups A and B, indicating no significant difference in beta diversity in the participants' microbiota when evaluated according to the post-menopausal period. For the Bray-Curtis and Jaccard indices (Fig 4A and 4B), there were no significant differences in the women's microbiota according to time in post-menopause.

### 4. Relative abundance of taxa inferred for bacteria in the fecal microbiota of post-menopausal women

Taxonomic inference indicated the presence of 14 phyla, 20 classes, 42 orders, 69 families, 180 genera, and 155 probable species in fecal samples from all study participants (n = 44) (Fig 5). Taxonomic inference data for the groups in this study are also presented (Table 3).

The 10 most abundant microorganisms of the intestinal microbiota were listed according to the relative frequency presented, and the Mann-Whitney test (U test) was performed to check whether there was a difference between these taxa between Groups A and B. The dominant phyla for groups A and B, respectively, were Firmicutes (58% and 66%),

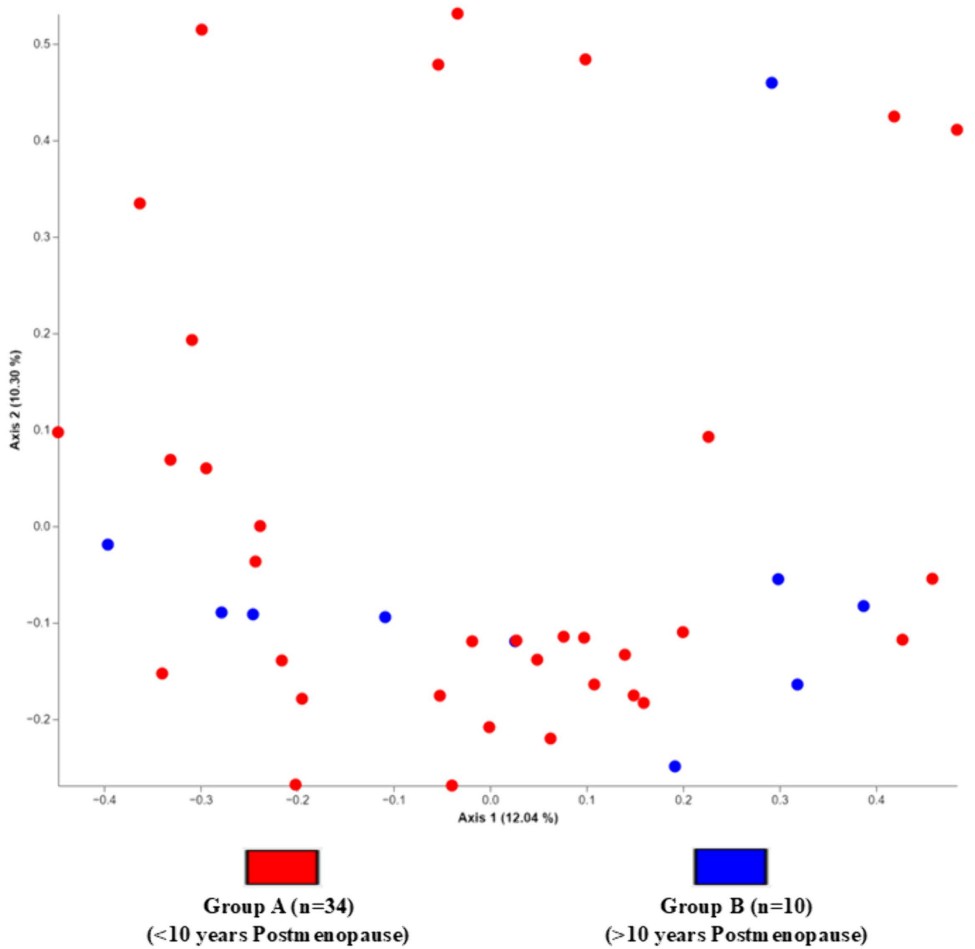

**Fig 3. Unweighted principal coordinate analysis (PCoA) of the bacterial communities presents in the fecal Microbiota of post-menopausal women.** Legend: Unweighted principal coordinate analysis (PCoA). Group A n = 34 (≤10 years postmenopause); Group B n = 10 (≥10 years postmenopause). The proportion of data variability described by the axes is 12.04% and 10.30%. Elaborated by the author.

Bacteroidota (36% and 28%), Proteobacteria (3% and 2.7%) and Actinobacteria (0.7% and 1.6%), these phyla jointly represent a proportion of 90% of the intestinal microbiota identified in each group (Fig 6).

The most abundant Families (Fig 7) for groups A and B respectively were: *Lachnospiraceae* (21.5% and 24%), *Bacteroidaceae* (19% and 18%), *Ruminococcaceae* (15% and 19%), *Prevotellaceae* (12% and 2%), *Veillonellaceae* (4.8% and 2.8%), *Oscillospiraceae* (4% and 5%), *Christensenellaceae* (2% and 1%), *Coprostanoligenes_group* and *Rikenellaceae* below 3% in both groups. The most abundant genera (Fig 8) for groups A and B, respectively, comprise *Bacteroides* (19% and 18%), *Prevotella* (10% and 1.5%), *Faecalibacterium* (8% and 9.6%), *Agathobacter* (4% and 4.7%), *Roseburia* (2.5% and 4%). The genera *Dialister, Subdoligranulum, Christensenellaceae_R-7_group*, and UCG-002 showed an abundance lower than 3% in both groups. No difference was found between the most enriched taxa when comparing the two groups in this study.

## Discussion

Female aging entails numerous changes, and estrogen decline is widely cited as one of the main hormonal manifestations in postmenopause, among others that also involve follicle-stimulating hormone (FSH), luteinizing

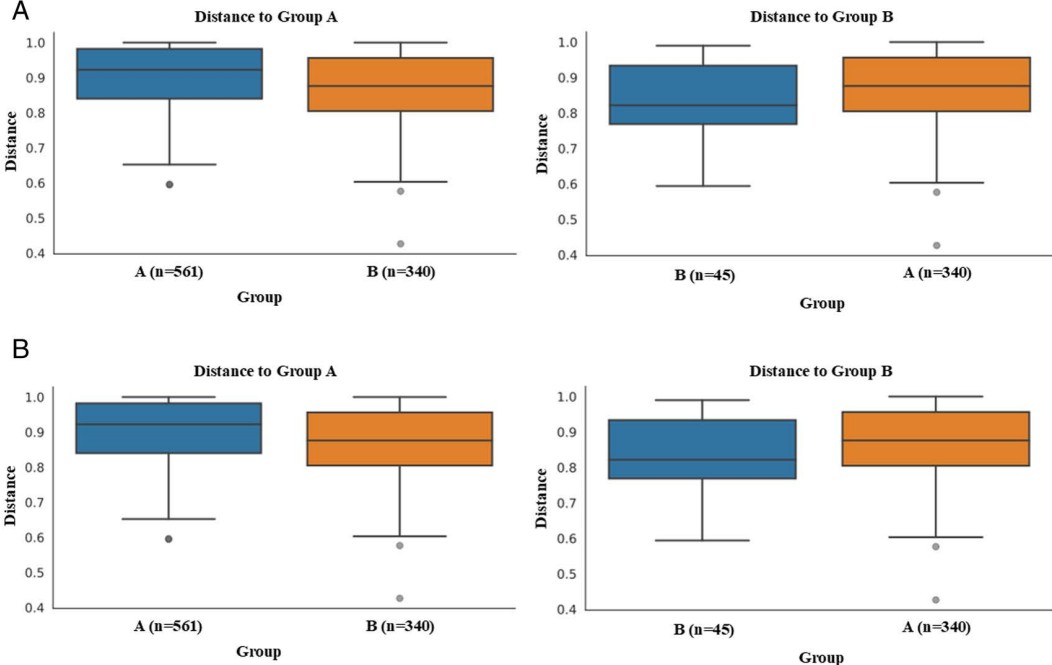

**Fig 4. Analysis of the fecal Microbiota's beta diversity according to postmenopause time. A)** Bray-Curtis distance using the ANOSIM test; **B)** Bray-Curtis distance using the PERMANOVA test. Legend: Group A n = 34 (≤10 years postmenopause); Group B n = 10 (≥10 years postmenopause). **A)** The ANOSIM (test R = −0.171; p-value = 0.98; N.Perm = 999). **B)** The statistical significance was assessed using permutational multivariate analysis of variance (PERMANOVA) (Pseudo-F = 0.972; p-value:0.492; N.Perm = 999). Elaborated by the author.

hormone (LH), and anti-Mulerian hormone (AMH) [5,43]. These changes, combined with lifestyle, genetics, socio-economic, and psychosocial conditions, act together in the manifestation of psychological, somatovegetative, and urogenital symptoms in the transition period from menopause to postmenopause [9,10]. In addition to these changes, aging leads to an adaptive change in the intestinal Microbiota [44,45], which, added to the decline in estrogen, can lead to a lower diversity of species and also changes in hormone reabsorption by the bacterial strobile [39,40].

This study investigated the composition and diversity of the gut microbiota of Brazilian women according to their time in postmenopause. In Brazil, few studies correlate gut microbiota and climacteric; the closest to the subject evaluated the impact of physical distancing due to COVID-19 on the composition of the gut microbiota of older men and women [41]. As far as our research group has investigated, this is the first Brazilian study to characterize the gut microbiota according to how long a woman has been living under post-menopausal conditions.

The study cohort consisted of 44 post-menopausal women, divided into those who had been post-menopausal for up to 10 years (Group A n = 34) and those who had been post-menopausal for more than 10 years (Group B n = 10), who had a mean age of 54.7 and 57.3 years respectively, the expected age range for Brazilian post-menopausal women. Conditions such as hypertension (A = 38% and B = 30%), diabetes (A = 20.6% and B = 20%), thyroid, bone and neuropsychiatric diseases (Table 1) were taken into consideration. There is a relationship between chronological aging and the development of metabolic disorders such as changes in glucose metabolism, insulin resistance, cardiovascular alterations, dyslipidemia, and obesity, as well as emotional, psychological, cognitive, and musculoskeletal alterations [15,42–45]. In addition to aging, the hormonal decline resulting from menopause has also been described as being associated with the onset of these conditions [16,46,47].

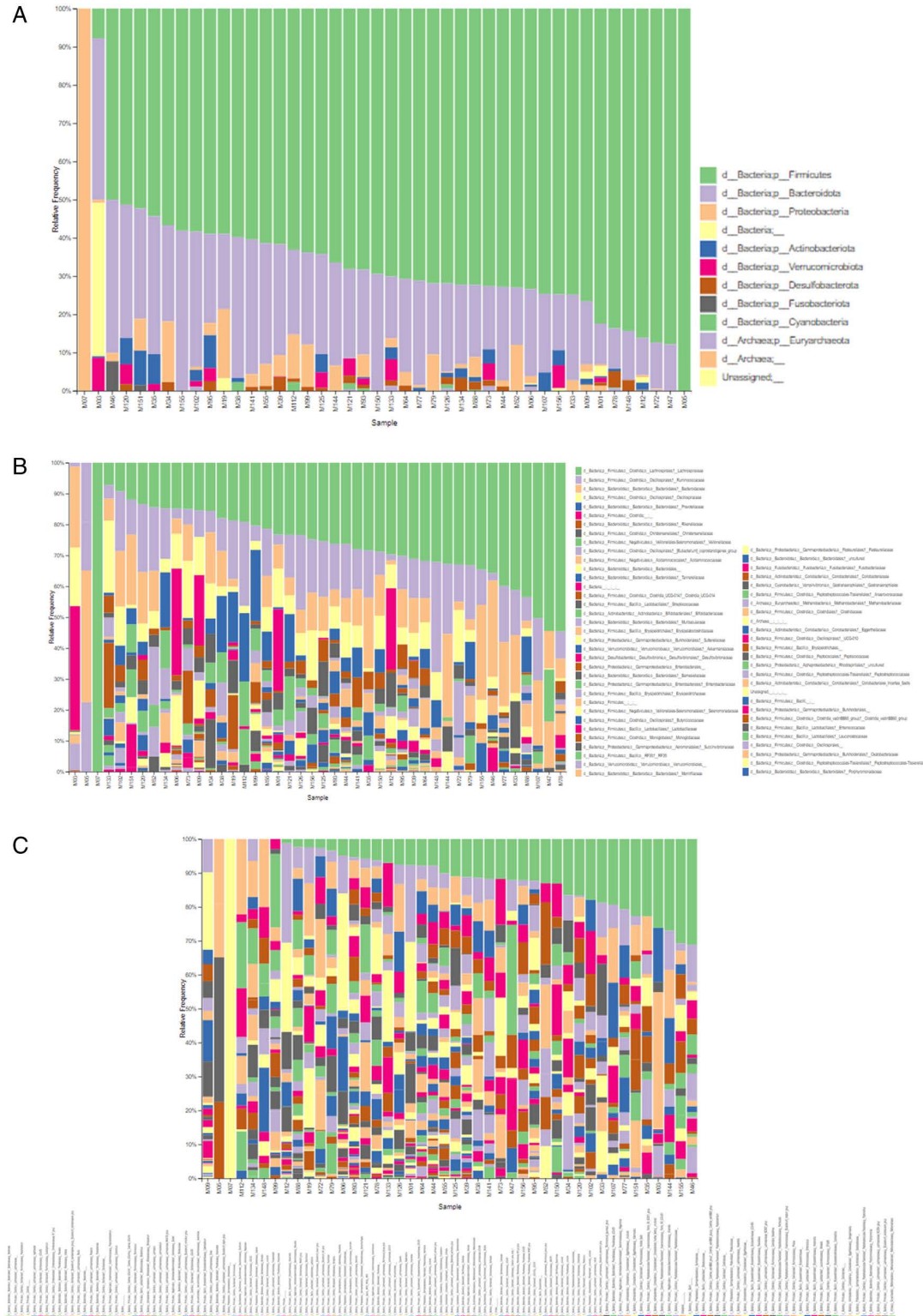

**Fig 5. Relative abundance analysis at phylum level (A), family level (B), and genus level (C) based on 16S rRNA data. A)** Phylum; **B)** Family; **C)** Genus. Predominant taxa as determined by 16S rRNA gene sequencing: a) Phylum, b) Family, c) Genus; The visual variations, although not quantitatively supported, may suggest microbial changes or trends specific to everyone that warrant further investigation. The samples are grouped by condition.

Group A n = 34 (≤10 years postmenopause) and Group B n = 10 (≥10 years postmenopause). Each column represents a sample, and different colors represent different relative abundances. Prepared by the author.

High-throughput sequencing used in marker gene studies has expanded rapidly in the last decade, particularly in identifying the taxonomic profile of microbial populations. In this study, the sequencing carried out on 44 samples after the filtering and quality control process obtained a total of 3.747.166 good quality reads (Table 2), and to achieve this result, the assembly method used was ASV (Amplicon Sequence Variant) [48,49]. The use of ASVs has been replacing methods based on OTUs (Operating Taxonomic Units), as they have advantages such as being reusable in other studies, reproducible in future data sets, and not suffering from the limitations of incomplete reference databases [48,50]. The high-throughput sequencing presented in this study successfully characterized the diversity and microbial communities present in the gut microbiota of post-menopausal women since the rarefaction curves (Fig 1) reached a plateau, demonstrating that the depth of the sequencing was satisfactory for capturing the taxonomic diversity present in the samples evaluated [51].

Using different diversity metrics to assess the intestinal Microbiota according to the length of time the participating women had been post-menopausal, the Shannon Index showed that both groups had high diversity, but there was no statistical difference between them (Fig 2A). Simpson's Index (Fig 2B) indicated that Group B (more than 10 years post-menopausal) had a bacterial community with greater diversity and lower taxon dominance, making it a more equitable bacterial community.

The gut microbiota of post-menopausal women tends to be less diverse than that of premenopausal women [39,52]. Our study showed that the post-menopausal time factor did not significantly affect bacterial diversity in the cohort evaluated since aging is already considered a key factor contributing to changes in intestinal bacterial communities [53,54].

In addition to women's chronological aging, the hormonal decline maintained in the post-menopausal state modifies Alpha diversity, leading to a less diverse microbiota. This mechanism is not yet fully understood [55]. Alpha and beta diversity analyses are essential tools for understanding the "invisible life" in our bodies. Alpha diversity analyses how many different species (or types of microorganisms) exist in a single place, such as a person's intestine [33,36,37]. It helps to understand whether that environment has a large variety or is dominated by a few species. Beta diversity, on the other hand, assesses the differences between two environments or groups, such as the intestines of healthy people compared to those of people with some disease [34,38]. Together, they allow inferences to be made: How much variety we have in the intestine (or other places) and how the Microbiota of one person or group compares to another. The Alpha and β-Diversity results found in this study's cohort indicate similarity between the organisms found in the fecal Microbiota of these women and a uniform and equitable distribution of the microorganisms inferred for the groups regardless of the time elapsed since postmenopause.

The 16S rRNA profile of the participants evaluated showed that the predominant phyla in both groups are Firmicutes, Bacteroidota, Proteobacteria, and Actinobacteria (Fig 6). Among the most abundant Families (Fig 7) are *Lachnospiraceae*, *Bacteroidaceae*, *Ruminococcaceae*, *Prevotellaceae*, *Veillonellaceae*, *Oscillospiraceae*, *Christensenellaceae*, *coprostanoligenes_group* and *Rikenellaceae*. The most abundant genera (Fig 8) were *Bacteroides*, *Prevotella*, *Faecalibacterium*, *Agathobacter*, *Roseburia*, and the genera *Dialister, Subdoligranulum, Christensenellaceae_R-7_group* and *UCG-002* were less than 3% abundant in both groups. The ten most abundant phyla in our cohort represented 90% of the entire microbiota within each group. We found a higher abundance of Families belonging to the predominant Phylum Firmicutes in this cohort and a lower abundance of Families belonging to *Bacteroidetes*, as seen previously [24,56,57].

Our study found no significant differences when evaluating the ten most abundant taxa for Phylum, Family, and Genus (Figs 6–8) in the study cohort. Post-menopausal women have estrogen levels decreased by up to 80% [1,44], and the bidirectional relationship that occurs between gut bacteria and the hormonal fluctuation that begins in menopause and

**Table 3. Taxonomic inference was found in each group according to the time after menopause.**

| Samples | Phylum | Class | Order | Family | Genus | Putative Species |
|---|---|---|---|---|---|---|
| **Total** (n = 44) | 14 (100%) | 20 (100%) | 42 (100%) | 69 (100%) | 180 (100%) | 155 (100%) |
| **GROUP A (n = 34)** (≤10 years postmenopause) | 14 (100%) | 20 (100%) | 41 (98%) | 68 (99%) | 171 (95%) | 150 (97%) |
| Group B (n = 10) (≥10 years postmenopause) | 10 (71%) | 15 (75%) | 34 (81%) | 55 (80%) | 154 (86%) | 95 (61%) |

For each taxonomic level, the actual number of taxa found and their relative abundance (%), the frequency that each taxonomic level presented about the total number of taxa found is presented. Elaborated by the author.

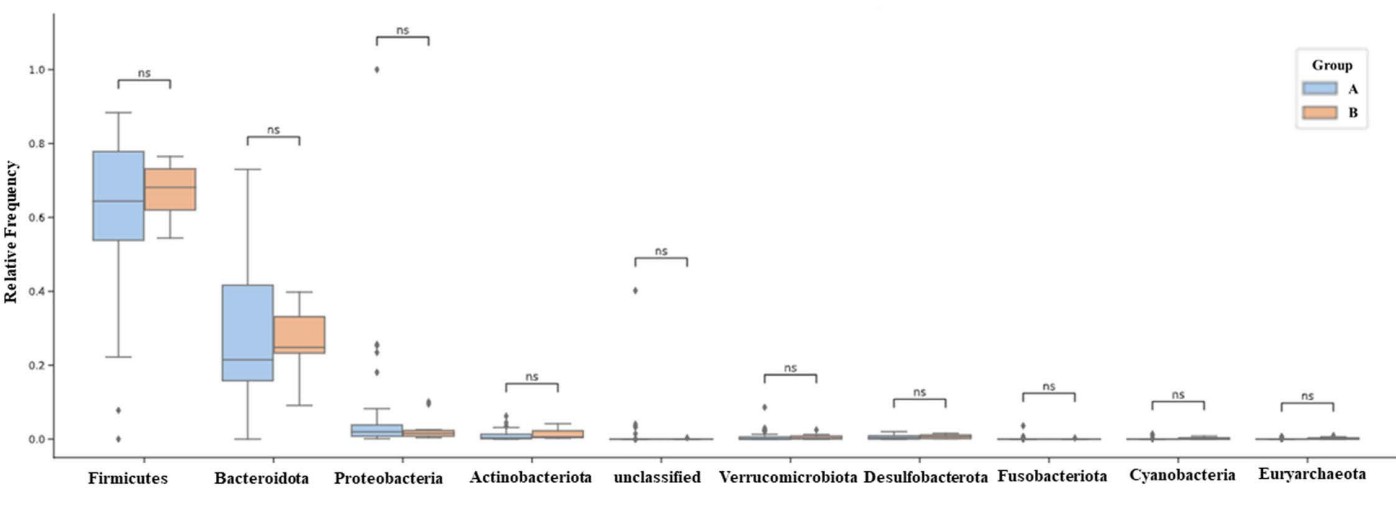

**Fig 6. Classification of the 10 most abundant phyla according to their relative abundance by groups.** Predominant phyla determined by 16S rRNA gene sequencing. Predominant bacterial phyla in the fecal microbiota of groups A and B, were determined by sequencing the 16S rRNA gene. Taxonomic classification was carried out using the DADA2 pipeline, with attribution based on the SILVA 138 reference database. The bars represent the average relative abundance for each group Group A n = 34 (≤10 years postmenopause) and Group B n = 10 (≥10 years postmenopause). The 10 main phyla are shown, with *Firmicutes*, *Bacteroidota*, *Proteobacteria* and *Actinobacteriota* representing more than 80% of the community composition in all groups. Data is expressed as relative frequency (%). Statistical analysis was carried out using the Mann-Whitney U-test. Each column represents a group. Data presented as relative frequency. Ns: Not significant.

stabilizes in postmenopause has been discussed [39,58]. Finding a similar intestinal microbiota in this cohort regardless of the years in postmenopause suggests that these participants have what has been called a resilient microbiota, which is the property that this microbial ecosystem has in recovering and maintaining its functional or taxonomic composition in the face of some disturbance [59].

The available studies investigating intestinal microbiota during a woman's life phases tend to compare the microbiota of premenopausal women with those of post-menopausal women [24–26,52,60]. The occurrence of female ovarian aging and the change from pre- to post-menopause is well understood due to hormonal changes, which mainly involve estrogen. There is evidence of bidirectional communication between the Sex Hormone-Intestinal Microbiome axis, which can modify circulating hormone levels and intestinal bacteria, including the estrobolome [39,58,60].

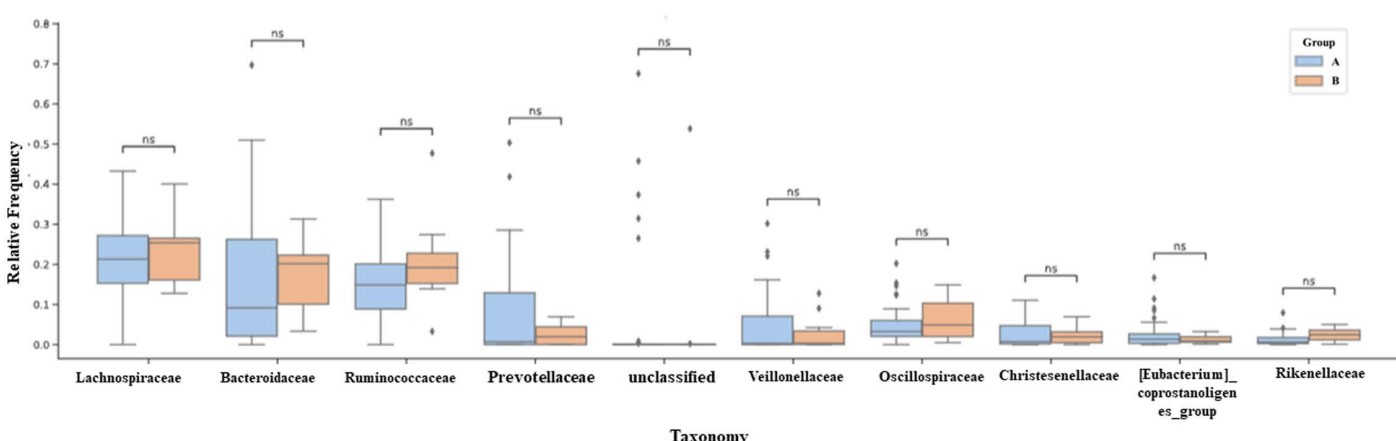

**Fig 7. Classify the 10 most abundant families according to their relative abundance by group.** Predominant families determined by 16S rRNA gene sequencing. Predominant bacterial families in the fecal microbiota of groups A and B, were determined by sequencing the 16S rRNA gene. Taxonomic classification was carried out using the DADA2 pipeline, with attribution based on the SILVA 138 reference database. The bars represent the average relative abundance for each group Group A n = 34 (≤10 years postmenopause) and Group B n = 10 (≥10 years postmenopause). The 10 main families are shown, with *Lachnospiraceae*, *Bacteroidaceae*, *Ruminococcaceae* and *Prevotellaceae* representing more than 80% of the community composition in all groups. Data is expressed as relative frequency (%). Statistical analysis was carried out using the Mann-Whitney U-test. Each column represents a group. Data presented as relative frequency. Ns: Not significant.

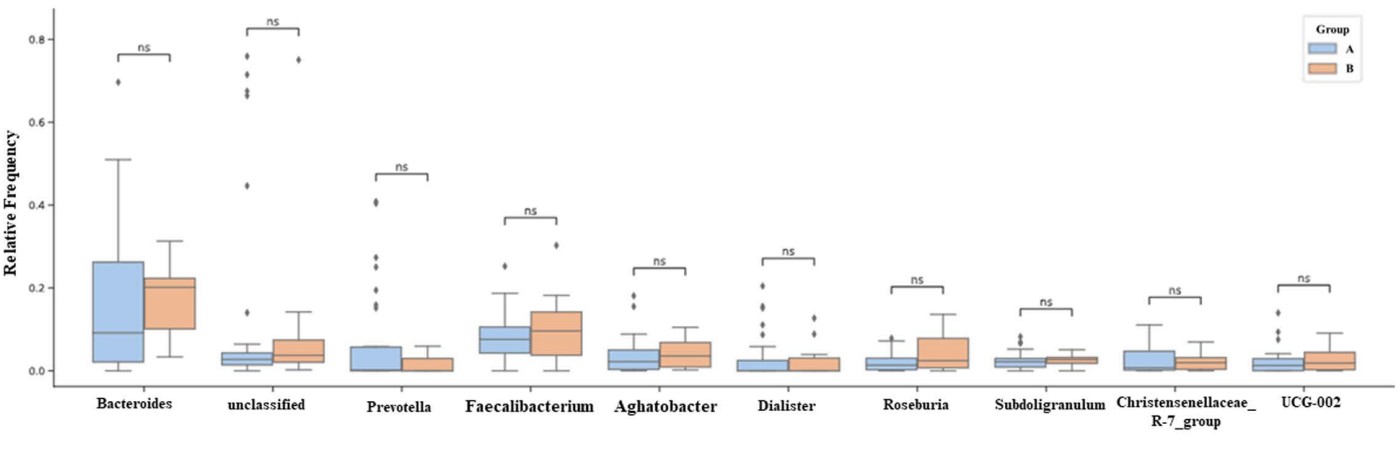

**Fig 8. Classification of the 10 most abundant genera according to their relative abundance by groups.** Predominant genera determined by 16S rRNA gene sequencing. Predominant bacterial genera in the fecal microbiota of groups A and B, were determined by sequencing the 16S rRNA gene. Taxonomic classification was carried out using the DADA2 pipeline, with attribution based on the SILVA 138 reference database. The bars represent the average relative abundance for each group Group A n = 34 (≤10 years postmenopause) and Group B n = 10 (≥10 years postmenopause). The 10 main genera are shown, with *Bacteroides*, *unclassified*, *Prevotella* and *Faecalibacterium* representing more than 80% of the community composition in all groups. Data is expressed as relative frequency (%). Statistical analysis was carried out using the Mann-Whitney U-test. Each column represents a group. Data presented as relative frequency. Ns: Not significant.

Information on the intestinal microbiome of the brazilian population is scarce since studies with large cohorts have been carried out in European, Asian, and North American countries [61]. However, the results presented for the women in this study about the most abundant phyla are in line with what has been found in the general population [62], in pre- and

post-menopausal women [24], and in older adults of both sexes [41]. Our study also corroborates Fuhrman *et al.* (2014), who evaluated only post-menopausal participants, indicating a higher relative abundance of Firmicutes followed by Bacteroidetes in their gut microbiota. There is also agreement with a study that showed Firmicutes inversely associated with low estradiol levels in women [56].

Numerous disturbances expose the microbiome throughout its host's life. The type of diet consumed, physical exercise or a sedentary lifestyle, infections, and the use of medication, particularly antibiotics, are widely reported as disturbances that can lead to unhealthy states of the microbiome and, depending on the degree of this bacterial disturbance, can be referred to as dysbiosis [63,64].

Women in our study reported concomitant use of more than one class of drugs for multiple conditions (Table 1). The use of drugs can also be considered a disturbance to the intestinal microbiota, modifying its diversity and even causing dysbiosis [65,66]. Pharmacomicrobiomics shows the bidirectional effect between medicines and the microbiota, and it is already known that oral medication can be absorbed by these bacteria, causing a reduction in their availability and, consequently, not achieving the desired effect [66,67]. Other medications absorbed by the microbiota may favor the increase of some species by modifying their diversity, for example, in the case of metformin [68–70], a drug used by some participants in our cohort.

Critical reviews show that the use of antidiabetic and antihypertensive drugs acts to modify and regulate the intestinal microbiota [64,70–72]. Firmicutes and Proteobacteria are enriched with the use of Captopril [65]. At the same time, Metformin is primarily related to the increase in genera such as *Prevotella* [68] and the enrichment of *Akkermansia muciniphila*, which reflects beneficial mechanisms such as the production of short-chain fatty acids by this bacterium, regulation of glycemic indices and protection and integrity of the gastrointestinal tract [73]. In contrast, Metformin reduced the population of *Roseburia* and *Faecalibacterium* [74]. Antidepressants also affect intestinal bacteria, modifying the concentration of secondary metabolites responsible for gut-brain communication [75].

Based on the literature already published, our research group recognizes the importance and relevance of the impact of drugs such as antihypertensives and antidiabetics on the composition of the gut microbiome. In this Brazilian cohort, these factors were recorded as potential confounding factors. Still, they were not directly included in the adjusted analyses due to the limited data stratified by specific drugs. For future studies, we recommend including a control group with similar characteristics but without medications and additional analyses to assess the possible influences of these medications on microbial diversity. The authors say that this unhealthy state of the microbiota can be transitory and return to a healthy level or move towards a fixed unhealthy state, which would be detrimental to the host's health [59].

A pilot study conducted in our laboratory in 2024 (unpublished data) used qPCR to evaluate the intestinal microbiota of post-menopausal women using hypoglycemic, antihypertensive, and lipid-lowering medications and those not using such medications. The genera *Akkermansia, Roseburia*, and *Faecalibacterium prausnitzii* were quantified, and no differences were found in the quantification of these organisms among the participants. This pilot study corroborates the sequencing findings (Fig 8), which also showed no differences for *Roseburia* and *Faecalibacterium prausnitzii*, indicating that such medications did not cause significant changes in these organisms in this cohort.

The hormonal changes women face when they enter menopause are considered a disturbance to their intestinal microbiota [25,56]. When reproductive aging sets in and post-menopause, the microbial core is established and adapts to the new environmental conditions the female organism provides, including less bacterial diversity and changes to the strobile [26,76]. Based on the results obtained by identifying the microbiota and their corresponding metrics, our research group strongly believes that the similarity found between the groups in this study can be explained and supported by the concept of resilient, resistant, and stable microbiota. Once a woman is post-menopausal, she does not return to the previous hormonal scenario, except in cases where hormone replacement therapy is used, which still minimizes the lack of estrogen but does not entirely restore its levels.

Since a woman will live a third of her life under the physiological conditions inherent to post-menopause, aging and hormonal decline can be considered host-derived disturbances [76]. Over time, these conditions become inherent to the woman's life phase, leading to an adaptation of these intestinal bacterial communities. Resilience is not static but a dynamic response to these environmental, age-related conditions [77].

Our research has significant limitations that must be reported when studying gut microbiota. Different sequencing methods (paired-end for 38 samples and single-end for 6) could bias the results. However, to minimize these effects, we applied standardized bioinformatics pipelines and normalized the data before analysis to make the reads comparable without prejudicing subsequent analyses.

We recognize the limitation of a smaller sample size (n = 10) in the post-menopausal group for more than 10 years. The small sample size reduces the statistical power to identify subtle differences in microbial composition. Future studies with more participants in this category will be essential to validate our initial observations and give greater robustness to the conclusions. It was impossible to investigate and monitor dietary habits and practice physical activities. The participants reported using different classes of medication, except antibiotics. We cannot rule out the possibility that these factors may influence the composition of these participants' microbiota. Future studies should be carried out to control these factors in this cohort, and predictive functional analysis should also be included to strengthen the characterization of the intestinal microbiota of Brazilian women.

The strengths of our study involve the participation of a cohort of post-menopausal women whose age range coincides with significant hormonal and metabolic changes primarily related to essential modifications in the intestinal microbiota. Our study contributes to the characterization of this microbiota in middle-aged Brazilian women, showing their resilience, which may open the way for individualized and personalized treatments for conditions that affect women at this stage of life. In addition, we characterized the ten most abundant organisms in this population, which could indicate new actions to prevent post-menopausal health conditions.

## Conclusion

Our study characterized the intestinal microbiota of post-menopausal women using high-throughput sequencing. According to Simpson's index, post-menopausal women who had been post-menopausal for ten years or more had a microbiota with greater diversity and lower species dominance when compared to post-menopausal women who had been post-menopausal for less than ten years. No difference existed between the ten most enriched organisms in each group at the Phylum, Family, and Genus taxonomic levels. Our results indicated that the fecal microbiota of these women had a uniform and equitable distribution of organisms inferred for the groups regardless of the time elapsed since postmenopause.

## Supporting information

**S1 Table. Correlation analysis between the ten most abundant bacterial phyla, families, and genera with clinical parameters (Group A n = 34 ≤ 10 YEARS POSTMENOPAUSE).** **Pearson's correlation with significant association with clinical parameters in post-menopausal women for less than ten years. Parametric data was tested using the Shapiro-Wilk test. **Spearman correlation with significant association with clinical parameters in post-menopausal women for less than ten years. Non-parametric data was tested using the Shapiro-Wilk test.
(DOCX)

**S2 Table. Correlation analysis between the ten most abundant bacterial phyla, families, and genera with clinical parameters (Group B n = 10 ≥ 10 YEARS POSTMENOPAUSE).** **Pearson's correlation with significant association with clinical parameters in post-menopausal women for more than ten years. Parametric data was tested using the

Shapiro-Wilk test. ** Spearman's correlation with significant association with clinical parameters in post-menopausal women for more than ten years. Parametric data was tested using the Shapiro-Wilk test.
(DOCX)

## Author contributions

**Conceptualization:** Aristóteles Góes-Neto, Vasco Ariston de Carvalho Azevedo, Renata Guerra-Sá.

**Data curation:** Thayane Christine Alves da Silva, Jennefer Aparecida Gonçalves Oliveira, Lauro Ângelo Gonçalves de Moraes, Izinara Rosse.

**Formal analysis:** Thayane Christine Alves da Silva, Lauro Ângelo Gonçalves de Moraes, Izinara Rosse, Aristóteles Góes-Neto, Vasco Ariston de Carvalho Azevedo, Renata Guerra-Sá.

**Funding acquisition:** Aristóteles Góes-Neto, Renata Guerra-Sá.

**Investigation:** Thayane Christine Alves da Silva, Jennefer Aparecida Gonçalves Oliveira.

**Methodology:** Thayane Christine Alves da Silva, Jennefer Aparecida Gonçalves Oliveira, Lauro Ângelo Gonçalves de Moraes, Izinara Rosse.

**Resources:** Aristóteles Góes-Neto, Vasco Ariston de Carvalho Azevedo, Renata Guerra-Sá.

**Supervision:** Aristóteles Góes-Neto, Vasco Ariston de Carvalho Azevedo, Renata Guerra-Sá.

**Validation:** Thayane Christine Alves da Silva, Renata Guerra-Sá.

**Visualization:** Thayane Christine Alves da Silva, Renata Guerra-Sá.

**Writing – original draft:** Thayane Christine Alves da Silva, Renata Guerra-Sá.

**Writing – review & editing:** Thayane Christine Alves da Silva, Renata Guerra-Sá.

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
