## [Decision Letter · Decision Letter 0]

8 Jan 2025

PONE-D-24-55161Microbiota and Postmenopause: The resilience of intestinal bacteria in the face of female hormonal aging.PLOS ONE

Dear Dr. Guerra-Sá,

Thank you for submitting your manuscript to PLOS ONE. After careful consideration, we feel that it has merit but does not fully meet PLOS ONE’s publication criteria as it currently stands. Therefore, we invite you to submit a revised version of the manuscript that addresses the points raised during the review process.

We look forward to receiving your revised manuscript.

Kind regards,

Sayed Haidar Abbas Raza

Academic Editor

PLOS ONE

Journal requirements: When submitting your revision, we need you to address these additional requirements. 1. Please ensure that your manuscript meets PLOS ONE's style requirements, including those for file naming. The PLOS ONE style templates can be found at https://journals.plos.org/plosone/s/file?id=wjVg/PLOSOne_formatting_sample_main_body.pdf and https://journals.plos.org/plosone/s/file?id=ba62/PLOSOne_formatting_sample_title_authors_affiliations.pdf. 2. Please amend either the title on the online submission form (via Edit Submission) or the title in the manuscript so that they are identical. 3. We note that the grant information you provided in the ‘Funding Information’ and ‘Financial Disclosure’ sections do not match.  When you resubmit, please ensure that you provide the correct grant numbers for the awards you received for your study in the ‘Funding Information’ section. 4. Thank you for stating the following financial disclosure:  [This research is supported by CAPES (Coordenação de Aperfeiçoamento de Pessoal de Nível Superior), CNPQ (Conselho Nacional de Desenvolvimento Científico e Tecnológico), Rede Minas Microbioma, Projeto RED-00181-23 FAPEMIG, and the Brazilian National Council for Scientific and Technological Development.].  Please state what role the funders took in the study.  If the funders had no role, please state: ""The funders had no role in study design, data collection and analysis, decision to publish, or preparation of the manuscript."" If this statement is not correct you must amend it as needed. Please include this amended Role of Funder statement in your cover letter; we will change the online submission form on your behalf. 5. We note that you have indicated that there are restrictions to data sharing for this study. PLOS only allows data to be available upon request if there are legal or ethical restrictions on sharing data publicly. For more information on unacceptable data access restrictions, please see http://journals.plos.org/plosone/s/data-availability#loc-unacceptable-data-access-restrictions.  Before we proceed with your manuscript, please address the following prompts: a) If there are ethical or legal restrictions on sharing a de-identified data set, please explain them in detail (e.g., data contain potentially identifying or sensitive patient information, data are owned by a third-party organization, etc.) and who has imposed them (e.g., a Research Ethics Committee or Institutional Review Board, etc.). Please also provide contact information for a data access committee, ethics committee, or other institutional body to which data requests may be sent. b) If there are no restrictions, please upload the minimal anonymized data set necessary to replicate your study findings to a stable, public repository and provide us with the relevant URLs, DOIs, or accession numbers. For a list of recommended repositories, please see https://journals.plos.org/plosone/s/recommended-repositories. You also have the option of uploading the data as Supporting Information files, but we would recommend depositing data directly to a data repository if possible. We will update your Data Availability statement on your behalf to reflect the information you provide. 6. In the online submission form, you indicated that [The data is available. Contact corresponding author Renata Guerra-Sá email rguerra@ufop.edu.br]. All PLOS journals now require all data underlying the findings described in their manuscript to be freely available to other researchers, either 1. In a public repository, 2. Within the manuscript itself, or 3. Uploaded as supplementary information.This policy applies to all data except where public deposition would breach compliance with the protocol approved by your research ethics board. If your data cannot be made publicly available for ethical or legal reasons (e.g., public availability would compromise patient privacy), please explain your reasons on resubmission and your exemption request will be escalated for approval.  7. Please include captions for your Supporting Information files at the end of your manuscript, and update any in-text citations to match accordingly. Please see our Supporting Information guidelines for more information: http://journals.plos.org/plosone/s/supporting-information. 

Reviewers' comments:

Reviewer's Responses to Questions

**Comments to the Author**

1. Is the manuscript technically sound, and do the data support the conclusions?

Reviewer #1: No

Reviewer #2: Yes

2. Has the statistical analysis been performed appropriately and rigorously? 

Reviewer #1: No

Reviewer #2: Yes

3. Have the authors made all data underlying the findings in their manuscript fully available?

Reviewer #1: Yes

Reviewer #2: Yes

4. Is the manuscript presented in an intelligible fashion and written in standard English?

Reviewer #1: Yes

Reviewer #2: Yes

5. Review Comments to the Author

Reviewer #1: This article provides an insightful look into how the gut microbiota of Brazilian women behaves after menopause and whether that composition varies according to how long a woman has been postmenopausal. By applying 16S rRNA gene sequencing, the authors highlight the taxonomic profile and diversity of key bacterial groups in postmenopausal women, showing overall stability and resilience in the microbiome. The study is especially valuable in filling a gap in Brazilian cohort data, where such large-scale microbiome assessments are relatively scarce. The findings support the notion that once a woman enters the postmenopausal phase, her microbiome may maintain equilibrium over time, despite hormonal and age-related changes.

Major Mistakes

1. While the article notes that some participants used antihypertensive, antidiabetic, and other medications, there is no in-depth discussion of how these confounders might have influenced the gut microbiota composition. More elaboration on how such factors were controlled or factored into the analysis would strengthen the study.

2.With only 10 women in the group that has been postmenopausal for more than 10 years, the statistical power to detect meaningful differences is somewhat limited.This may question whether sample size restrictions impacted the study’s ability to detect subtler microbiome shifts.

3. While alpha and beta diversity metrics and certain statistical tests (Kruskal-Wallis, PERMANOVA, etc.) are mentioned, the text does not elaborate on multiple testing corrections or whether the p-values were adjusted for multiple comparisons. Clarifying these points would add rigor to the findings.

Minor Mistakes

1. Occasional double “Note: Note:” statements and extra spaces around parentheses can distract readers. A careful proofreading for typographical errors (e.g., extraneous spacing, repeated words) would improve readability.

2. The manuscript mentions figures and tables (Figure 3, Table 2, etc.) that are presumably provided, but some figure or table references might appear out of order or without a fully explanatory caption. Ensuring the figures and tables are clearly labeled, cited in the correct order, and have complete legends will help the flow of the text.

3. In a few paragraphs, ideas about aging, hormone decline, and microbiota changes are repeated in very similar wording. More concise phrasing or combining those sentences under the same subheading would avoid redundancy.

4. Some terms like “p-value,” “F-value,” or how certain tests were run (e.g., “999 permutations”) could be formatted consistently across all results sections. A small but notable improvement would be using standard statistical reporting guidelines uniformly.

Reviewer #2: Major:

The study does not adequately control for potential confounding factors such as diet, physical activity, and medication use. These factors can significantly influence the gut microbiome composition. The authors should address how these variables were accounted for in their analysis or discuss them as limitations.

The sequencing methods differed for some samples (38 paired-end vs. 6 single-end). The authors should explain how this discrepancy might affect the results and whether any measures were taken to mitigate potential biases.

The authors should consider using more robust statistical techniques for microbiome data analysis, such as PERMANOVA or ANOSIM for beta diversity comparisons.

The study focuses primarily on taxonomic composition but lacks insight into the functional implications of the observed microbiome differences. The authors should consider incorporating predictive functional profiling tools to provide more context to their findings.

Minor:

Authors need to include a more detailed description of the participants' demographic and clinical characteristics in the results section, possibly as a table.

Authors need to provide more information on the DNA extraction and library preparation protocols, including any quality control measures.

Authors need to include a brief explanation of the chosen alpha and beta diversity metrics for readers less familiar with microbiome analyses.

Authors need to consider adding a correlation analysis between specific bacterial taxa and relevant clinical parameters (e.g., time since menopause, age, BMI) to provide more context to the findings.

6. PLOS authors have the option to publish the peer review history of their article (what does this mean? ). If published, this will include your full peer review and any attached files.

**Do you want your identity to be public for this peer review?** For information about this choice, including consent withdrawal, please see our Privacy Policy .

Reviewer #1: No

Reviewer #2: No

---

## [Author Response · Author response to Decision Letter 1]

20 Feb 2025

The manuscript was checked for PLOS ONE style requirements, titles, subtitles, symbols and file nomenclature.

2. Please amend either the title on the online submission form (via Edit Submission) or the title in the manuscript so that they are identical.

The title on the online submission form and the manuscript have been revised.

The Financing and Financial Disclosure information has been revised.

4. Thank you for stating the following financial disclosure: [This research is supported by CAPES (Coordenação de Aperfeiçoamento de Pessoal de Nível Superior), CNPQ (Conselho Nacional de Desenvolvimento Científico e Tecnológico), Rede Minas Microbioma, Projeto RED-00132-16 FAPEMIG (Fundação de amparo a pesquisa Minas Gerais). Please state what role the funders took in the study. If the funders had no role, please state: "The funders had no role in study design, data collection and analysis, decision to publish, or preparation of the manuscript." If this statement is not correct you must amend it as needed. Please include this amended Role of Funder statement in your cover letter; we will change the online submission form on your behalf.

This study was funded in part by CAPES (Coordination for the Improvement of Higher Education Personnel - Brazil) - Funding code 001. By the Rede Minas Microbiome, Project RED-00132-16 FAPEMIG (Fundação de amparo a pesquisa de Minas Gerais) and by CNPq (Conselho Nacional de Desenvolvimento Científico e Tecnológico).

5. We note that you have indicated that there are restrictions to data sharing for this study. PLOS only allows data to be available upon request if there are legal or ethical restrictions on sharing data publicly.

b) If there are no restrictions, please upload the minimal anonymized data set necessary to replicate your study findings to a stable, public repository and provide us with the relevant URLs, DOIs, or accession numbers. For a list of recommended repositories, please see https://journals.plos.org/plosone/s/recommended-repositories. You also have the option of uploading the data as Supporting Information files, but we would recommend depositing data directly to a data repository if possible.

A dataset and workflow can be found at the following public domain address: https://github.com/bioinfonupeb/redemicro-thayane

The sequencing data is deposited at NCBI under the BioProject ID PRJNA1223961 http://www.ncbi.nlm.nih.gov/bioproject/1223961

6. In the online submission form, you indicated that [The data is available. Contact corresponding author Renata Guerra-Sá email rguerra@ufop.edu.br]. All PLOS journals now require all data underlying the findings described in their manuscript to be freely available to other researchers, either 1. In a public repository, 2. Within the manuscript itself, or 3. Uploaded as supplementary information. This policy applies to all data except where public deposition would breach compliance with the protocol approved by your research ethics board. If your data cannot be made publicly available for ethical or legal reasons (e.g., public availability would compromise patient privacy), please explain your reasons on resubmission and your exemption request will be escalated for approval.

A dataset and workflow can be found at the following public domain address: https://github.com/bioinfonupeb/redemicro-thayane

The sequencing data is deposited at NCBI under the BioProject ID PRJNA1223961 http://www.ncbi.nlm.nih.gov/bioproject/1223961

In addition to this, other important data has been made available as supplementary information.

7. Please include captions for your Supporting Information files at the end of your manuscript, and update any in-text citations to match accordingly. Please see our Supporting Information guidelines for more information:

Captions have been included in the supporting information files, figures and tables. The order of citation in the text has also been checked.

ANSWER TO REVIEWER n°1

1. While the article notes that some participants used antihypertensive, antidiabetic, and other medications, there is no in-depth discussion of how these confounders might have influenced the gut microbiota composition. More elaboration on how such factors were controlled or factored into the analysis would strengthen the study.

Answer: We acknowledge the importance of antihypertensive and antidiabetic drugs in shaping gut microbiome composition. While these factors were recorded as potential confounders, they were not directly included in the adjusted analyses due to the limited availability of data stratified by specific medications. For future studies, we recommend incorporating a control group with similar characteristics but without medication use, as well as conducting additional analyses to better assess the potential impact of these drugs on microbial diversity.

Please see lines 478-493: Critical reviews show that the use of antidiabetic and antihypertensive drugs acts to modify and regulate the intestinal microbiota (65,71,72). Firmicutes and Proteobacteria are enriched with the use of Captopril (65). At the same time, metformin is primarily related to the increase in genera such as Prevotella (68) and the enrichment of Akkermansia muciniphila, which reflects beneficial mechanisms such as the production of short-chain fatty acids by this bacterium, regulation of glycemic indices and protection and integrity of the gastrointestinal tract (73). In contrast, metformin reduced the population of Roseburia and Faecalibacterium (74). Antidepressants also affect intestinal bacteria, modifying the concentration of secondary metabolites responsible for gut-brain communication (75).

Based on the literature already published, our research group recognizes the importance and relevance of the impact of drugs such as antihypertensives and antidiabetics on the composition of the gut microbiome. In this Brazilian cohort, these factors were recorded as potential confounding factors. Still, they were not directly included in the adjusted analyses due to the limited data stratified by specific drugs. For future studies, we recommend including a control group with similar characteristics but without medications and additional analyses to assess the possible influences of these medications on microbial diversity.

Please see lines 521-539: Our research has significant limitations that must be reported when studying gut microbiota. We recognize the limitation of a smaller sample size (n=10) in the postmenopausal group for more than 10 years. The small sample size reduces the statistical power to identify subtle differences in microbial composition. Future studies with more participants in this category will be essential to validate our initial observations and give greater robustness to the conclusions. It was not possible to investigate and monitor dietary habits and practice of physical activities. The participants reported using different classes of medication, except antibiotics. We cannot rule out the possibility that these factors may influence the composition of these participants' microbiota. Future studies should be carried out to control these factors in this cohort, and predictive functional analysis should also be included to strengthen the characterization of the intestinal microbiota of Brazilian women.

The strengths of our study involve the participation of a cohort of postmenopausal women whose age range coincides with significant hormonal and metabolic changes primarily related to essential modifications in the intestinal microbiota. Our study contributes to the characterization of this microbiota in middle-aged Brazilian women, showing their resilience, which may open the way for individualized and personalized treatments for conditions that affect women at this stage of life. In addition, we characterized the ten most abundant organisms in this population, which could indicate new actions to prevent postmenopausal health conditions.

2.With only 10 women in the group that have been postmenopausal for more than 10 years, the statistical power to detect meaningful differences is somewhat limited. This may question whether sample size restrictions impacted the study’s ability to detect subtler microbiome shifts.

Answer: We acknowledge the limitation of a smaller sample size (n=10) in the group of postmenopausal women for more than 10 years, which may reduce the statistical power to detect subtle differences in microbial composition. However, despite this limitation, our results remain relevant, as there are few studies focusing on Brazilian cohorts. Future studies with a larger number of participants in this category will be essential to validate our initial findings and strengthen the robustness of our conclusions.

Please see lines 521-539: Our research has significant limitations that must be reported when studying gut microbiota. We recognize the limitation of a smaller sample size (n=10) in the postmenopausal group for more than 10 years. The small sample size reduces the statistical power to identify subtle differences in microbial composition. Future studies with more participants in this category will be essential to validate our initial observations and give greater robustness to the conclusions. It was not possible to investigate and monitor dietary habits and practice of physical activities. The participants reported using different classes of medication, except antibiotics. We cannot rule out the possibility that these factors may influence the composition of these participants' microbiota. Future studies should be carried out to control these factors in this cohort, and predictive functional analysis should also be included to strengthen the characterization of the intestinal microbiota of Brazilian women.

The strengths of our study involve the participation of a cohort of postmenopausal women whose age range coincides with significant hormonal and metabolic changes primarily related to essential modifications in the intestinal microbiota. Our study contributes to the characterization of this microbiota in middle-aged Brazilian women, showing their resilience, which may open the way for individualized and personalized treatments for conditions that affect women at this stage of life. In addition, we characterized the ten most abundant organisms in this population, which could indicate new actions to prevent postmenopausal health conditions.

3. While alpha and beta diversity metrics and specific statistical tests (Kruskal-Wallis, PERMANOVA, etc.) are mentioned, the text does not elaborate on multiple testing corrections or whether the p-values were adjusted for multiple comparisons. Clarifying these points would add rigor to the findings.

Answer: Although alpha and beta diversity analyses were conducted using PERMANOVA and Kruskal-Wallis tests, we acknowledge the importance of explicitly addressing multiple testing corrections. To mitigate false positives, we will apply standard corrections, such as the Benjamini-Hochberg method, and clarify these details in the text to enhance transparency and methodological rigor.

Please see lines 164-165: The Benjamini-Hochberg correction method was used to adjust the p-values.

Minor Mistakes

1. Occasional double “Note: Note:” statements and extra spaces around parentheses can distract readers. A careful proofreading for typographical errors (e.g., extraneous spacing, repeated words) would improve readability.

Answer: We thank you for your comment and will review the manuscript thoroughly to correct typographical errors, such as repetitions of 'Note: Note:' and unnecessary spacing. This effort will also include eliminating redundancies in the sections related to aging, hormonal decline, and changes in the microbiome, consolidating the information for better clarity.

Please see lines 01-931: All text has been revised

2. The manuscript mentions figures and tables (Figure 3, Table 2, etc.) that are presumably provided, but some figure or table references might appear out of order or without a fully explanatory caption. Ensuring the figures and tables are clearly labeled, cited in the correct order, and have complete legends will help the flow of the text.

Answer: We thoroughly checked the figures and tables to ensure that they were all cited in the correct order and with full explanatory captions. This improvement will help maintain the flow of the text and make it easier for the reader to understand. All figures and tables are clearly labeled, with a title and explanatory caption, and cited in the correct order in the text.

Please see lines 182-188: Table 1. Sociodemographic and health characteristics of the women participating in this study.

Please see lines 200-203: Table 2. Data inferred after quality control and merging reads obtained for single-end and paired-end.

Please see lines 311-315: Table 3. Taxonomic inference was found in each group according to the time after menopause.

Please see lines 205-209: Fig 1. Rarefaction curve of the features observed per sample analyzed (n=44).

Please see lines 224-240: Fig 2. Alpha diversity of the fecal microbiota presented by the Shannon and Simpson index.

Please see lines 248-258: Fig 3. Unweighted principal coordinate analysis (PCoA) of the bacterial communities presents in the fecal microbiota of postmenopausal women.

Please see lines 260-278: Fig 4. Analysis of the fecal microbiota's beta diversity according to post menopause time.

Please see lines 281-301: Fig 5. Relative abundance analysis at Phylum level (a), Family level (b) and Genus level (c) based on 16S rRNA data.

Please see lines 335-342: Fig 6. Classification of the 10 most abundant phyla according to their relative abundance by groups.

Please see lines 344-351: Fig 7. Classification of the 10 most abundant Family according to their relative abundance by groups.

Please see lines 353-359: Fig 8. Classification of the 10 most abundant Genera according to their relative abundance by groups.

Please see lines 952-996 Supplementary material: S1 Table. Medications used by participants in this study.

Please see lines 952-996 Supplementary material: S2 Table. Correlation analysis between the ten most abundant bacterial phyla, families and genera with clinical parameters

3. In a few paragraphs, ideas about aging, hormone decline, and microbiota changes are repeated in very similar wording. More concise phrasing or combining those sentences under the same subheading would avoid redundancy.

Answer: Thank you for your comment. We have reviewed the terms indicated and modified them for a more concise reading.

4. Some terms like “p-value,” “F-value,” or how certain tests were run (e.g., “999 permutations”) could be formatted consistently across all results sections. A small but notable improvement would be using standard statistical reporting guidelines uniformly.

Answer: We agree that consistency in formatting statistical terms, such as 'p-value' and 'F-value,' is fundamental. We will adjust to following standardized statistical reporting guidelines, using uniform formatting throughout the manuscript.

Please see lines 275-278: Legend: Group A n=34 (≤10 years postmenopause); Group B n=10 (≥10 years postmenopause). A) The ANOSIM (test R = -0.171; p-value = 0.98; N.Perm = 999). B) The statistical significance was assessed using permutational multivariate analysis of variance

---

## [Decision Letter · Decision Letter 1]

3 Mar 2025

PONE-D-24-55161R1Microbiota and Postmenopause: The resilience of intestinal bacteria in the face of female hormonal aging.PLOS ONE

Dear Dr. Guerra-Sá,

Thank you for submitting your manuscript to PLOS ONE. After careful consideration, we feel that it has merit but does not fully meet PLOS ONE’s publication criteria as it currently stands. Therefore, we invite you to submit a revised version of the manuscript that addresses the points raised during the review process.

We look forward to receiving your revised manuscript.

Kind regards,

Sayed Haidar Abbas Raza

Academic Editor

PLOS ONE

Journal Requirements:

Reviewers' comments:

Reviewer's Responses to Questions

**Comments to the Author**

1. If the authors have adequately addressed your comments raised in a previous round of review and you feel that this manuscript is now acceptable for publication, you may indicate that here to bypass the “Comments to the Author” section, enter your conflict of interest statement in the “Confidential to Editor” section, and submit your "Accept" recommendation.

Reviewer #1: All comments have been addressed

Reviewer #2: All comments have been addressed

2. Is the manuscript technically sound, and do the data support the conclusions?

Reviewer #1: Yes

Reviewer #2: Yes

3. Has the statistical analysis been performed appropriately and rigorously? 

Reviewer #1: Yes

Reviewer #2: Yes

4. Have the authors made all data underlying the findings in their manuscript fully available?

Reviewer #1: Yes

Reviewer #2: Yes

5. Is the manuscript presented in an intelligible fashion and written in standard English?

Reviewer #1: Yes

Reviewer #2: Yes

6. Review Comments to the Author

Reviewer #1: The revised manuscript now addresses all the previous concerns with clear, comprehensive improvements. The changes meet the publication standards, and I recommend accepting the paper for publication.

Reviewer #2: Concerns:

Certain paragraphs reiterate similar points without adding new insights.

The readers could benefit from a table summarizing participants’ BMI, comorbidities, and medication use.

The authors are suggested to add the different biasis with the different sequencing method.

Readers would be also benefitted from mentioning of functional implications of microbiome changes.

7. PLOS authors have the option to publish the peer review history of their article (what does this mean? ). If published, this will include your full peer review and any attached files.

**Do you want your identity to be public for this peer review?** For information about this choice, including consent withdrawal, please see our Privacy Policy .

Reviewer #1: **Yes: ** Simna Saraswathi Prasannakumari

Reviewer #2: No

---

## [Author Response · Author response to Decision Letter 2]

8 Apr 2025

Journal Requirements:

The list of references has been fully revised as requested. All references have been supplied correctly. We do not cite references that have been retracted.

ANSWER TO REVIEWER n°2

1. Certain paragraphs reiterate similar points without adding new insights.

Answer: We appreciate your comment and have carefully revised the manuscript to minimize similar points and make it more straightforward. As it was not specified, we marked them in the text, and you can check the changes in the paragraphs and lines where we removed and added them.

2. The readers could benefit from a table summarizing participants' BMI, comorbidities, and medication use.

Answer: Thank you for your suggestion. The information requested on the participants' BMI, comorbidities, and medication use has been added to Table 1.

Please see lines 191- 196: Table 1. Sociodemographic and health characteristics of the women participating in this study.

3. The authors are suggested to add the different biasis with the different sequencing method.

Answer: We acknowledge that using different sequencing methods (paired-end for 38 samples and single-end for 6) could introduce biases in the results. We applied standardized bioinformatics pipelines to minimize these effects and normalized the data before analysis. We have included this information as a limitation of our study, clarifying the potential impact on the results by making the methodology bias transparent to readers.

Please see lines 520- 523: Different sequencing methods (paired-end for 38 samples and single-end for 6) could bias the results. However, to minimize these effects, we applied standardized bioinformatics pipelines and normalized the data before analysis to make the reads comparable without prejudicing subsequent analyses.

4. Readers would be also benefitted from mentioning of functional implications of microbiome changes.

Answer: Thank you for your suggestion. Our research group would like to highlight a few important points based on a recent study done by Turjeman and collaborators (Turjeman S, Rozera T, Elinav E, Ianiro G, Koren O. From big data and experimental models to clinical trials: Iterative strategies in microbiome research. Cell. 2025 Mar 6;188(5):1178-1197. doi:10.1016/j.cell.2025.01.038. PMID: 40054445.)

Studies involving the intestinal microbiome face many challenges. One of the most significant challenges is the functional implications of changes in intestinal bacterial communities. The authors discuss at length that the field of research studying the microbiome and human diseases has been based mainly on correlative studies, and determining, for example, whether dysbiosis is a cause or consequence of diseases and their adjacent mechanisms is still the biggest challenge.

Finding correlations does not mean causality, as described by the authors. Extensive studies that have demonstrated some relationship between microbial modifications and the occurrence of diseases such as inflammatory bowel disease, colorectal cancer, diabetes, and host metabolism have yet to provide direct proof of this causal relationship between the microbiome and clinical outcomes. Even with advanced modeling and integrated multi-omics, this correlation and validation remain in silico.

Our research group agrees that for more compelling statements about the functional implications of modifications to the human microbiome, it is necessary to employ an iterative approach combining in silico, in vitro, ex vivo, and in vivo data, as suggested by the aforementioned work. For this reason, we have chosen not to speculate on functional mechanisms in this work. We intend to apply this iterative approach to the continuity of our future research with this cohort.

We believe we have answered all the reviewer's concerns. The changes have greatly improved the final version of the manuscript, for which we thank our colleagues' scientific assistance. In its current form, we hope that our manuscript is suited for publication in Plos One. Please advise us if any other changes are necessary.

Cordially,

Prof. Dr. Renata Guerra-Sá

(Corresponding Author)

Laboratory of Biochemistry and Molecular Biology (LBBM), Department of Biological Sciences, Institute of Exact and Biological Sciences, Federal University of Ouro Preto (UFOP). Campus Universitário s/n, Morro do Cruzeiro, Ouro Preto, MG, Brazil.

Postal code: 35400-000. E-mail: rguerra@ufop.edu.br

---

## [Decision Letter · Decision Letter 2]

30 Apr 2025

Microbiota and Postmenopause: The resilience of intestinal bacteria in the face of female hormonal aging.

PONE-D-24-55161R2

Dear Dr. Guerra-Sá,

We’re pleased to inform you that your manuscript has been judged scientifically suitable for publication and will be formally accepted for publication once it meets all outstanding technical requirements.

Kind regards,

Sayed Haidar Abbas Raza

Academic Editor

PLOS ONE

Additional Editor Comments (optional):

Reviewers' comments:

Reviewer's Responses to Questions

**Comments to the Author**

1. If the authors have adequately addressed your comments raised in a previous round of review and you feel that this manuscript is now acceptable for publication, you may indicate that here to bypass the “Comments to the Author” section, enter your conflict of interest statement in the “Confidential to Editor” section, and submit your "Accept" recommendation.

Reviewer #2: All comments have been addressed

2. Is the manuscript technically sound, and do the data support the conclusions?

Reviewer #2: Yes

3. Has the statistical analysis been performed appropriately and rigorously? 

Reviewer #2: Yes

4. Have the authors made all data underlying the findings in their manuscript fully available?

Reviewer #2: Yes

5. Is the manuscript presented in an intelligible fashion and written in standard English?

Reviewer #2: Yes

6. Review Comments to the Author

Reviewer #2: The authors have addressed all the comments previousl ythat wre suggested to them.Although they have address all the authors are suggested to insert images as higher fomat becuase in teh current state the figures look stretched and blurred.

7. PLOS authors have the option to publish the peer review history of their article (what does this mean? ). If published, this will include your full peer review and any attached files.

**Do you want your identity to be public for this peer review?** For information about this choice, including consent withdrawal, please see our Privacy Policy .

Reviewer #2: No

---

## [Editor Report · Acceptance letter]

PONE-D-24-55161R2

PLOS ONE

Dear Dr. R,

I'm pleased to inform you that your manuscript has been deemed suitable for publication in PLOS ONE. Congratulations! Your manuscript is now being handed over to our production team.

Kind regards,

on behalf of

Dr. Sayed Haidar Abbas Raza

Academic Editor

PLOS ONE